# Long Term Follow-Up Study of a Randomized, Open-Label, Uncontrolled, Phase I/II Study to Assess the Safety and Immunogenicity of Intramuscular and Intradermal Doses of COVID-19 DNA Vaccine (AG0302-COVID19)

**DOI:** 10.3390/vaccines11101535

**Published:** 2023-09-28

**Authors:** Hironori Nakagami, Tetsuya Matsumoto, Kenji Takazawa, Hisakuni Sekino, Osamu Matsuoka, Satoshi Inoue, Hidetoshi Furuie, Ryuichi Morishita

**Affiliations:** 1Department of Health Development and Medicine, Osaka University Graduate School of Medicine, 2-2 Yamada-oka, Suita 565-0871, Japan; 2Department of Infectious Diseases, Graduate School of Medicine, International University of Health and Welfare, Narita Hospital, 852 Hatakeda Narita, Chiba 286-0124, Japan; tetsuya.m@iuhw.ac.jp; 3Medical Corporation Shinanokai Shinanozaka Clinic, 20 Samon-cho, Shinjuku-ku, Tokyo 160-0017, Japan; 4Sekino Clinical Pharmacology Clinic, 3-28-3 Ikebukuro, Toshima-Ku, Tokyo 171-0014, Japan; 5Medical Corporation Heishinkai ToCROM Clinic, 4-9, Yotsuyasanei-cho, Shinjuku-ku, Tokyo 160-0008, Japan; 6Medical Corporation Heishinkai OCROM Clinic, 4-12-11, Kasuga, Suita 565-0853, Japan; satoshi.inoue@heishinkai.com; 7Osaka Pharmacology Clinical Research Hospital, 4-1-29, Miyahara, Yodogawa-ku, Osaka 532-0003, Japan; hidetoshi.furuie@heishinkai.com; 8Department of Clinical Gene Therapy, Osaka University Graduate School of Medicine, 2-2 Yamada-oka, Suita 565-0871, Japan; morishit@cgt.med.osaka-u.ac.jp

**Keywords:** COVID-19, DNA vaccine, SARS-CoV-2

## Abstract

Pharmacological studies have demonstrated antibody production and infection prevention with an intradermal coronavirus disease 2019 (COVID-19) DNA vaccine (AG0302-COVID-19). This clinical trial aimed to investigate the safety and immunogenicity of high doses of AG0302-COVID19 when injected intramuscularly and intradermally. Healthy adults were randomly divided into three intramuscular vaccination groups (2 mg, three times at 2-week intervals; 4 mg, twice at 4-week intervals; and 8 mg, twice at 4-week intervals) and two intradermal groups (1 mg, three times at 2-week intervals or twice at 4-week intervals). After a one-year follow-up, no serious adverse events were related to AG0302-COVID-19. At Week 52, the changes in the geometric mean titer (GMT) ratios of the anti-S antibodies were 2.5, 2.4, and 3.2 in the 2, 4, and 8 mg intramuscular groups, respectively, and 3.2 and 5.1 in the three times and twice injected intradermal groups, respectively. The number of INF-γ-producing cells responsive to S protein increased after the first dose and was sustained for several months. AG0302-COVID-19 showed an acceptable safety profile, but the induction of a humoral immune response was insufficient to justify progressing to a Phase 3 program.

## 1. Introduction

Coronavirus disease 2019 (COVID-19) is an infectious disease caused by SARS-CoV-2. The main clinical symptoms of this disease include fever, respiratory symptoms, and general malaise; many patients present with mild symptoms, but some experience more serious issues such as dyspnea, which can lead to death when it turns to severe pneumonia. After the first infection was confirmed in Wuhan City, Hubei Province, People’s Republic of China, in December 2019, the disease spread across China and the whole world, following which COVID-19 was declared a “public health emergency of international concern” by the World Health Organization on 30 January 2020 [1].

Several treatment methods, including viral vector, attenuated, and mRNA vaccines, have been developed to prevent infection in patients with COVID-19 [2,3,4,5,6]. The DNA vaccine can be designed soon after the gene sequence is specified; moreover, it is possible to scale up the manufacturing process of this vaccine in Japan. Therefore, we started to develop a DNA vaccine with an excellent therapeutic effect on COVID-19 infection.

AG0302-COVID19 is a gene-drug created at Osaka University, in which the gene sequence of the spike (S) protein present on the surface of SARS-CoV-2 is incorporated into the plasmid backbone pVAX1^®^. In vitro and in vivo pharmacological studies have demonstrated the production of antibodies against AG0302-COVID19 and the infection prevention activities of this drug [7]. AG0302-COVID19 is being developed in humans via two routes of vaccination: intramuscular and intradermal. Intramuscular vaccination involves a mixture of AG0302-COVID19 and an aluminum phosphate adjuvant, whereas a specific device is used for the intradermal AG0302-COVID19 dose.

Phase I/II studies on intramuscular and intradermal AG0302-COVID19 vaccination, including the 2 mg/intramuscular and 0.2 and 0.4 mg/intradermal doses, have been conducted [8,9]. A Phase II/III study comprising a 2 mg/dose intramuscular inoculation was underway after confirming the immunogenicity of two intramuscular inoculations (2 mg/dose). However, it is currently possible to manufacture and supply concentrated preparations of AG0302-COVID19. Therefore, the aim of this clinical trial was to investigate the safety and immunogenicity of high doses of AG0302-COVID19 injected intramuscularly or intradermally.

## 2. Materials and Methods

### 2.1. Trial Design and Patients

The study protocol, contents of the informed consent document for the subjects, and propriety of conducting the study were approved by the IRB of the medical institution before implementation. This clinical trial complied with the protocol, “GCP”, “Declaration of Helsinki” and the necessary regulatory requirements, and was registered at the Clinical Trials. gov (Identifier: NCT04993586).

This study, conducted between 29 July 2021, and 23 September 2022, consisted of screening, vaccination, observation, and follow-up periods. After obtaining the informed consent, a screening test was conducted, and eligible subjects were registered. The subjects were randomly divided into two groups based on the vaccination route (intramuscular or intradermal). The number of cases was registered to avoid discrepancies in the administration methods and doses between the two groups. This study was conducted in the context of an approved public vaccination program to prevent COVID-19; hence, participation could be terminated prematurely at any time to allow for participation in the public vaccination program. The sponsor provided investigators and others with up-to-date information about the investigational drug at the time of briefing so that subjects could decide whether or not to participate in the public vaccination program.

The key inclusion criteria in this study included the following: 18 years of age or older, provision of a written informed consent to participate in the study, and negative for severe acute respiratory syndrome coronavirus 2 (SARS-CoV-2) via polymerase chain reaction (PCR) testing or negative for both SARS-CoV-2 IgM and SARS-CoV-2 IgG via antibody testing. The exclusion criteria were as follows: symptoms of suspected COVID-19 infection (e.g., respiratory symptoms, headaches, fatigue, olfactory disorders, and dysgeusia), a history of COVID-19 infection or vaccination with a vaccine (approved or unapproved) to prevent COVID-19, an axillary temperature of ≥37.5 °C at screening, an axillary temperature of ≥37.5 °C prior to the first dose of the study drug, pregnant or lactating female, and a history of international travel within 4 weeks prior to the start of dosing with the study drug.

### 2.2. Vaccine and Medical Device

AG0302-COVID-19 consists of a DNA plasmid vector, pVAX1, which carries genes that express SARS-CoV-2 spike S-proteins. Viral RNA for SARS-CoV-2 (Wuhan, Hu-1 isolate; MN_908,947.3) was obtained from the National Institute of Infectious Diseases (Tokyo, Japan). SARS-CoV-2, a highly optimized DNA sequence encoding a spike glycoprotein, was constructed using the in silico gene optimization algorithm to enhance protein expression [6]. Plasmid DNA was grown for large-scale production under current Good Manufacturing Practice conditions. The investigational drugs AG0302-COVID19 2 mg/mL (lot number: AG0302-DP-007) and AG0302-COVID19 4 mg/mL (lot number: AG0302-DP-008) were administered intramuscularly with an aluminum adjuvant and intradermally using a Pro-Drive Jet Injector [10].

### 2.3. Procedure

The subjects were divided into three intramuscular injection groups (Group A: 2 mg of plasmid DNA, three times at 2-week intervals; Group B: 4 mg of plasmid DNA, two times at 4-week intervals; and Group C: 8 mg of plasmid DNA, twice at 4-week intervals [both arms]), and two intradermal injection groups (Group D: 1 mg of plasmid DNA, three times at 2-week intervals and Group E: 1 mg of plasmid DNA twice at 4-week intervals), with a target of 80 patients in each group. In the intramuscular vaccination groups, safety was confirmed 5 days after the first injection of the study drug in at least five subjects in each group (the day of injection was defined as day 1). This was followed by the randomization of the subsequent subjects and the inoculation of the study drug. The same measures were taken for those in the intradermal vaccination groups. The intramuscular and intradermal vaccinations were performed in parallel. Observations, surveys, or tests were performed for 52 weeks after the first dosing.

### 2.4. Primary and Secondary Endpoints


**Primary endpoint**
Adverse events, adverse reactions, serious adverse events, and specific adverse events (local reactions, systemic reactions) that occurred between the time of the first dose and 12 weeks after the first dose of the study drug and up to 2 weeks after each dose.Neutralizing activity against SARS-CoV-2 pseudovirus (50% inhibitory concentration [ID_50_]) and SARS-CoV-2 spike (S) glycoprotein-specific antibody fold increase in GMT (GMT rate of change).



**Secondary endpoint**
Seroconversion rate of neutralizing activity (ID_50_) against SARS-CoV-2 pseudovirus: number and percentage of subjects with a ≥4-fold increase in neutralizing activity (ID_50_) against the SARS-CoV-2 pseudovirus.Changes in the number of interferon (IFN)-γ-producing cells after stimulation of peripheral blood mononuclear cells with SARS-CoV-2 spike (S) glycoprotein.Protective effect against infection: the proportion of subjects infected with SARS-CoV-2 from the first dose of the study drug to 52 weeks after the first dose (vaccination, observation, and follow-up periods).


### 2.5. Laboratory Analysis

The SARS-CoV-2 spike (S) glycoprotein-specific antibody titer, SARS-CoV-2 spike (S) glycoprotein-specific antibody IgG subclass (IgG1), neutralizing activity against SARS-CoV-2 pseudovirus were measured at Nexelis (Québec, QC, Canada), and IFN-γ production against SARS-CoV-2 spike (S) glycoprotein in peripheral blood mononuclear cells, which were measured at Cancer Precision Medicine (Tokyo, Japan).


**SARS-CoV-2 spike (S) glycoprotein**
**-specific antibody titer**


Antibody levels specific for the SARS-CoV-2 spike (S) glycoprotein in human sera were quantified by the enzyme-linked immunosorbent assay (ELISA). The SARS-CoV-2 spike (S) glycoprotein-specific antibody in captured human serum was detected using a horseradish peroxidase-conjugated goat anti-human polyclonal antibody with a SARS-CoV-2 spike (S) glycoprotein as a solid-phase antigen and was converted to a SARS-CoV-2 spike (S) glycoprotein specific antibody concentration (ELU/mL) using a standard curve. The minimum dilution ratio was 50; human serum was diluted in seven steps using a common ratio of 2. A cutoff value of 50.3 ELU/mL was used.


**SARS-CoV-2 Neutralizing activity against pseudovirus**


The virus-neutralizing activity in human sera was determined by measuring the neutralizing antibody titers using pseudovirus as an alternative to SARS-CoV-2. A pseudovirus consisting of replication-deficient VSV vector (VSVΔG-Luc) encoding luciferase, instead of the G protein in the bullous stomatitis virus (VSV-G), and expressing the SARS-CoV-2 spike(S) glycoprotein was used. Diluted human sera and the pseudovirus were incubated at 37 °C for 60 min before plating onto Vero E6 cells. The number of luciferase-positive infected cells was counted 18–22 h after inoculation and compared with the serum-free pseudovirus-positive control. The dilution ratio, at which the number of infected cells reached 50%, was calculated and used as the neutralizing antibody titer. The minimum dilution ratio was set to 10.


**Production of IFN-γ**
**in SARS-CoV-2 spike (S) glycoprotein**
**-stimulated peripheral blood mononuclear cells.**


The number of IFN-γ-producing cells in SARS-CoV-2 spike (S) glycoprotein-stimulated human peripheral blood mononuclear cells was counted using the ELISPOT assay; 1.0 × 10^6^ human peripheral blood mononuclear cells were counted.


**PCR and antigen test**


PCR and antigen tests were performed at weeks 12, 24, and 52 to confirm infection with COVID-19.

### 2.6. Statistical Analysis

The changes in GMT for the antibody titer and neutralizing antibody at 2 (just before the second dose), 4, 6, 8, 12, 20, and 24 weeks after the first dose of the study drug and its two-sided 95% confidence interval (CI) were calculated overall and for each dose group, using the baseline GMT as the reference. In terms of the IFN-γ production, the summary statistics for the measurements were calculated using the same method at each time point. The number of subjects and the two-sided 95% CIs of the adverse events were calculated for the safety evaluation. If the 50 inhibitory dilution or ELISA titer was less than the cutoff value, its half the cutoff value was utilized for calculating GMT. The statistical analysis was performed using SAS version 9.4 (SAS Institute Inc., Cary, NC, USA). The Japanese version of the ICH International Medical Terminology (MedDRA/J) was used to replace the adverse events, the MedDRA/J system was used as the organ classification, and the preferred term was used as the name of the adverse event.

## 3. Results

### 3.1. Composition and Baseline Characteristics of the Subjects

A summary of the composition during the study is presented in Figure 1. A total of 567 subjects who provided signed informed consent for the study were screened, and 448 were randomized into five groups: 85 in Group A, 84 in Group B, 83 in Group C, 84 in Group D, and 86 in Group E. The subjects received the first vaccination between 5 August 2021, and 15 October 2021.

During the observation period, 11 subjects (3 from Group A, 2 from Group B, 1 from Groip C, 4 from Group D, and 1 from Group E) discontinued the study. Subsequently, 42 subjects (9 from Group A, 4 from Group B, 10 from Group C, 13 from Group D, and 6 from Group E) discontinued the study during the follow-up period. None of the subjects dropped out due to adverse events. The baseline demographics (sex, age, height, and weight), medical history, underlying disease, and smoking history were well-balanced among the five treatment groups (Table 1).

### 3.2. Safety and Tolerability

After the first dose of the study drug, and through the subsequent 12 weeks (vaccination and observation periods, respectively), serious adverse events (acute bile capsule inflammation, colon polyp, and acute pancreatitis) were observed in one subject in Group C. The events were confirmed as unrelated to the drug. At 12–52 weeks after the first dose of the drug (follow-up period), serious adverse events were encountered in two subjects in Group C (one with COVID-19 infection and one with spinal compression fracture) and two subjects in Group E (one with cholelithiasis and the other with adnexal torsion). The COVID-19 infection, cholelithiasis, and adnexal torsion were severe, while the spinal compression fracture was moderately severe. All these serious adverse events were unrelated to the investigational drug and resolved in due course. There were no adverse events leading to discontinuation of the study drug administration.

The incidence of adverse events, which occurred up to 12 weeks after each administration of the study drug, was 79.4% (216/252 subjects) in the intramuscular group and 84.7% (144/170 subjects) in the intradermal group (Table 2). The adverse events, which occurred in more than 5% of subjects in the intramuscular group, comprised pain (78.2%) and induration (11.5%) at the vaccination site, diarrhea (7.1%), malaise (21.0%), and headache (20.6%). The incidence of adverse events in the intradermal inoculation group comprised pain (40.6%), induration (31.8%), erythema (32.4%), pruritus (20.6%), and swelling (20.6%) at the vaccination sites, diarrhea (18.2%), malaise (30.6%), and headache (26.5%).

### 3.3. Immune Responses

SARS-CoV-2 spike (S) glycoprotein-specific antibody titers are shown in Figure 2. Log antibody titer at Week 12 was 1.67 ± 0.47 in Group A, 1.55 ± 0.32 in Group B, 1.61 ± 0.42 in Group C, 1.67 ± 0.47, and 1.55 ± 0.35 in Gropp E. The GMP ratio of SARS-CoV-2 spike (S) glycoprotein-specific antibodies at Week 12 to baseline was 1.4 (95% CI, 1.0 to 1.6) in Group A, 1.2 (95% CI, 1.0 to 1.3) in Group B, 1.7 (95% CI, 1.3 to 2.1) in Group C, 1.6 (95% CI, 1.3 to 2.0) in Group D, and 1.4 (95% CI, 1.2 to 1.7) in Group E. The GMT of the ELISA titer after vaccination remained in the <2-fold range during the observation period (12 weeks). The changes in the GMT ratios of the SARS-CoV-2 spike (S) glycoprotein-specific antibodies at Week 52 were 2.5 in Group A, 2.4 in Group B, 3.2 in Group C, 3.2 in Group D, and 5.1 in Group E.

The neutralizing activity (ID_50_) against the SARS-CoV-2 pseudovirus is shown in Figure 3. Log neutralizing activity at Week 12 was 0.74 ± 0.19 in Group A, 0.78 ± 0.32 in Group B, 0.78 ± 0.33 in Group C, 0.83 ± 0.31, and 0.77 ± 0.31 in Gropp E. The GMP ratio of neutralizing activity (ID_50_) against SARS-CoV-2 pseudovirus at Week 12 to baseline was 1.0 (95% CI, 1.0 to 1.1) in Group A, 1.1 (95% CI, 1.0 to 1.1) in Group B, 1.1 (95% CI, 1.0 to 1.3) in Group C, and 1.3 (95% CI, 1.0 to 1.4) in Group D. 1.1 (95% CI, 1.0 to 1.3) in group E. The GMP ratio of neutralizing activity (ID_50_) against SARS-CoV-2 pseudovirus at Week 12 to baseline was 1.0 (95% CI, 1.0 to 1.1) in Group A, 1.1 (95% CI, 1.0 to 1.1) in Group B, 1.1 (95% CI, 1.0 to 1.3) in Group C, and 1.3 (95% CI, 1.0 to 1.4) in Group D. 1.1 (95% CI, 1.0 to 1.3) in group E. The mean ID_50_ value remained in the <2-fold range during the observation period. At Week 52, the positive conversion rate for the ID_50_ against the SARS-CoV-2 pseudovirus, the proportion of subjects with at least a four-fold increase from before the first dose of the study drug, was 14.5% in Group A, 14.3% in Group B, 22.0% in Group C, 23.1% in Group D, and 24.4% in Group E.

Bars represent the mean and standard error values for the number of INF-γ-positive PBMCs after stimulation by the SARS-CoV-2 spike (S) glycoprotein.

Figure 4 shows the changes in the number of IFN-γ-producing cells induced by stimulation with SARS-CoV-2 spike (S) glycoprotein in peripheral blood mononuclear cells. The highest mean (standard deviation) numbers were 864 (706) in Group A at Week 12, 593 (581) in Group B at Week 8, 862 (845) in Group C at Week 4, 964 (841) in Group D at week 12, and 856 (685) in Group E at Week 8. Higher values were noticed in the groups with the administered dosage of every 2 weeks regardless of the route of vaccination.

None of the subjects were infected with SARS-CoV-2 during the observation period. However, the proportion of subjects infected with SARS-CoV-2 during the follow-up period was 18.9% (47/249) in the intramuscular vaccination group (13.3% in Group A); 20.5% in Group B); 22.9% in Group C), and 22.8% (38/167) in the intradermal vaccination group (20.7% in Group D and 24.7% in Group E).

## 4. Discussion

Various approaches have been taken to enhance the immune responses in clinical trials for application in humans. The optimization of plasmid DNA vectors (use of strong promoters/enhancers or increased adjuvant action by inserting CpG motifs) is important to increase the immunogenicity and potency of the gene expression. DNA vaccine can be delivered intramuscularly or intradermally, which primarily induces the expression of antigen in myocytes and keratinocytes, respectively, but also in antigen presenting cells near the injection side [11,12,13]. In addition, the delivery and transfection system has been optimized, and jet injectors [14], liposomes [15,16], and electroporation [17] have enhanced responses through increased efficiency of DNA delivery. Additionally, a gene-delivery system (intradermal injection, intracutaneous administration of microparticles with Pyro-drive jet injectors comprising a gas generator, piston, plunger, and container) was used for the rapid development of DNA vaccines in this study [10]. The formulation was optimized with aluminum salt [18], a clinically safe adjuvant that has been used clinically for several vaccines (e.g., diphtheria, tetanus, pertussis).

Twelve weeks after the first dose of the study drug, no apparent increase in GMTs for the neutralizing activity against the SARS-CoV-2 pseudovirus (ID_50_) or the SARS-CoV-2 spike (S) glycoprotein-specific antibodies was observed, regardless of the route of administration. These findings suggest that the DNA vaccine may not be as powerful as the mRNA vaccine in increasing the antibody titer at the early phase. During the long-term follow-up until 52 weeks, many subjects were infected with SARS-CoV-2 due to the spread of the mutant variant. The remaining subjects may have been exposed to SARS-CoV-2 infection, but were not infected in this study. Interestingly, the antibody titer was increased at 52 weeks in all groups, which may have prevented the spread of SARS-CoV-2 infection.

After vaccination, an increase in the number of IFN-γ-producing cells was observed among peripheral blood mononuclear cells stimulated with SARS-CoV-2 spike (S) glycoprotein in all five groups, indicating that the cell-mediated immunity was activated. The number of IFN-γ-producing cells demonstrated a slow decline after rising to a peak in association with a low but sustained level of the spike protein antibody, suggesting that the nuclear-incorporated plasmid may have persistently supplied the spike proteins. Bange et al. reported that COVID-19 patients with hematologic cancer who has COVID-19-specific INF-gamma response showed improved mortality even in limited humoral immunity condition [19], and they pointed out the importance of the CD8 T cell response to vaccination. In the future, we may be confronting highly transmissible and more pathogenic variants, therefore, the development of a technologies capable of inducing T cell immune responses based on the DNA vaccine may be a suitable way of preventing the progression to severe disease and death.

Drug delivery systems, administration devices (such as electroporation and needle-free syringes), administration routes, and adjuvants have been investigated to improve the immunization of the plasmid DNA, and efforts to increase gene transfer have been attempted to date. However, compared to other conventional vaccines, such as vector-type DNA vaccines, the DNA from plasmid vaccines may not be efficiently transduced into the cells. Although gene expression requires the translocation of cytoplasmic DNA to the nucleus, plasmid DNA generally has a low rate of nuclear translocation in non-proliferating cells [20]. The development of a new method to efficiently introduce foreign DNA into the nucleus in the future is warranted. Dose independency in production of IFN-γ- cells observed in this study might be partly affected by this inefficient penetration into cell nuclei. In addition, the route or the method of delivery did not have any differential effect on the antibody titers post vaccination between intramuscular and intradermal vaccination, suggesting no apparent difference in the sequential antigen synthesis process (i.e., transfection of plasmid DNA into antigen presenting cells, endogenous genes expression, antigen presentation via major histocompatibility in dendric cells, and T-cell responses).

As for safety profile, both local and systemic adverse events observed in this study were reported with other different types of vaccines, and no adverse events that were considered unique to this vaccine were reported. Although this is a rough comparison, it seemed that the incidence and severity of systemic adverse events, such as fatigue, headache, fever, and chills with this DNA vaccine, was lower than that of mRNA and vector-type vaccines [2,3,4,6], and does not much differ from that of the inactivated vaccine [5].

INOVIO in the United States and Zydus in India are developing DNA vaccines using the spike glycoprotein as an antigen [21]. INOVIO was among the first to announce the development of a DNA vaccine targeting COVID-19, using an electroporation system for intradermal administration (CELLECTRA^®^ 2000 device), which had already been researched and developed for other infectious diseases. In a Phase 1 study (NCT 4336410) conducted after evaluation in animal experiments, two doses of 1 mg and 2 mg were administered to 20 subjects at 4-week intervals for each dose [21]. Although an increase in the antibody and neutralizing antibody titer was confirmed at the time, the dose dependence was unclear. Almost no systemic adverse reactions and only local pain at the injection site were observed in a few patients. On the other hand, during the evaluation of the cell-mediated immunity, the expression of IFN-γ was increased in both groups; however, the increase was significantly higher in the 2 mg group compared to the 1 mg group [21]. Additionally, their study showed that the neutralizing activity after vaccination was lower than that of convalescent plasma, similar to that reported in clinical studies on other DNA vaccines [22,23], even though the potencies of vector and RNA vaccines are comparable to those of convalescent plasma. Zydus is conducting a clinical trial using the Biojector (needle-free syringe); in the Phase 1 trial, two doses of 1 mg and 2 mg were administered to 24 people at 4-week intervals for two doses [24]. In each dose group, 12 subjects underwent needle injection, and 12 underwent intradermal injection using a needleless syringe. Evaluations of the antibody and neutralizing antibody titers revealed favorable effects in the group that received the 2 mg needle-free administration. Zydus reported an efficacy of 66.6% in the preliminary report, and emergency approval was granted in India, making it the first DNA vaccine to be approved. However, it is likely that the current status of nonviral DNA vaccines may not show clear advantages over existing vaccines, such as mRNA and vector vaccines, in terms of antibody production [25].

## 5. Conclusions

Both intramuscular and intradermal injections of AG0302-COVID19 induced T cell immunity, but AG0302-COVID19 could not elicit sufficient humoral immunity for transition to Phase 3 in this study. No particular concerns about side effects were found. The advantages of DNA vaccines are that they can be produced easily, quickly, and on a large scale, are safer than other methods, such as inactivated virus vaccines, and are safer than other vaccines. Although COVID-19 has subsided, the technology for highly effective DNA vaccines must be expanded and marketed in preparation for the future.

## Figures and Tables

**Figure 1 vaccines-11-01535-f001:**
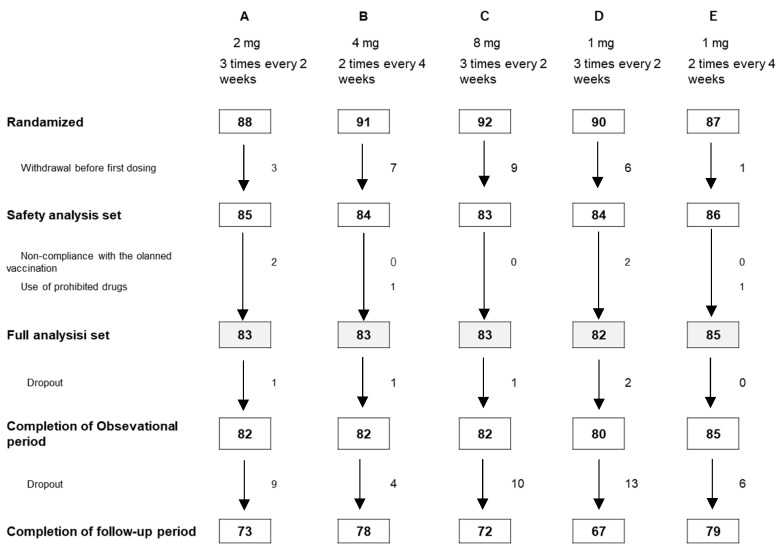
Enrollment, randomization, and withdrawal of the subjects during the study.

**Figure 2 vaccines-11-01535-f002:**
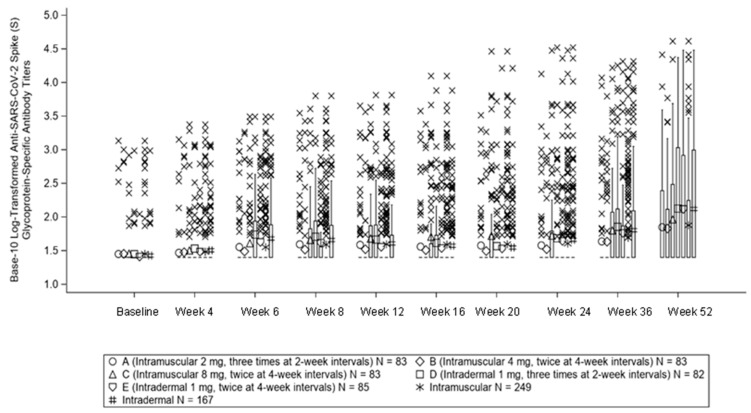
Box-Whisker plot of anti-SARS-CoV-2 spike (S) glycoprotein-specific antibody titers (Log_10_ ELISA titer).

**Figure 3 vaccines-11-01535-f003:**
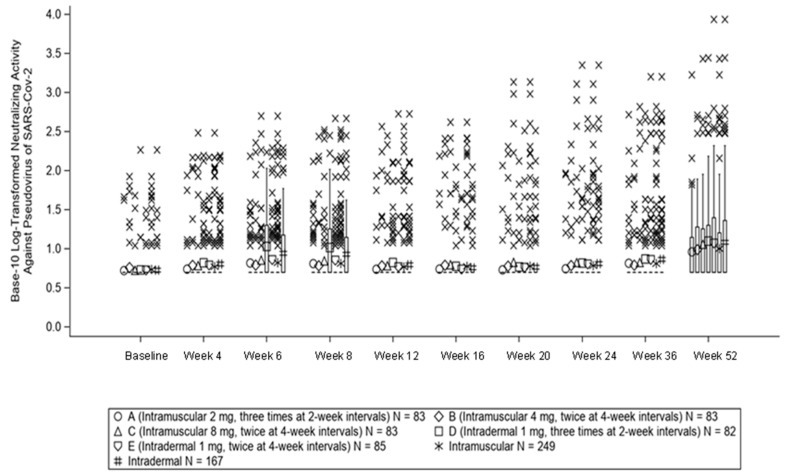
Box-Whisker plot of neutralizing activity (Log ID_50_) against pseudovirus of SARS-CoV-2.

**Figure 4 vaccines-11-01535-f004:**
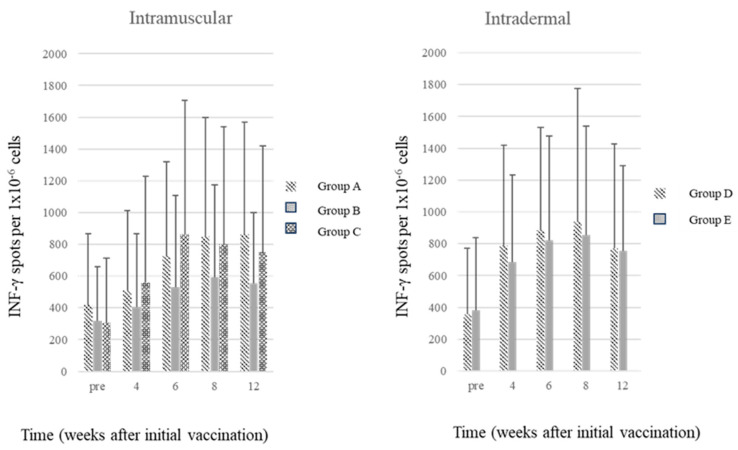
T-cell responses following AG0302-COVID19 vaccine in SARS-CoV-2-naive participants.

**Table 1 vaccines-11-01535-t001:** Demographics characteristics (Full analysis set).

		A	B	C	D	E
Demographics		(2 mg, Three Times at 2-Week Intervals) *n* = 83	(4 mg, Twice at 4-Week Intervals) *n* = 83	(8 mg, Twice at 4-Week Intervals) *n* = 83	(1 mg, Three Times at 2-Week Intervals) *n* = 82	(1 mg, Twice at 4-Week Intervals) *n* = 85
Sex	male	43	44	43	40	42
	Female	40	39	40	42	43
Age	Mean ± SD	49 ± 14	48 ± 15	49 ± 14	46 ± 13	47 ± 15
	Range	(21–77)	(20–75)	(22–79)	(21–80)	(20–79)
Height	Mean ± SD	165 ± 9	165 ± 9	164 ± 9	165 ± 9	164 ± 8
	Range	(146–185)	(148–183)	(145–179)	(142–188)	(149–181)
Weight	Mean ± SD	60 ± 11	62 ± 10	60 ± 12	61 ± 13	63 ± 13
	Range	(32–92)	(46–99)	(35–95)	(42–106)	(38–105)
Medical history	Yes	1	3	3	6	2
Underlying diseases	yes	52	60	53	63	70
Smoking history	past/current	16/8	10/12	16/7	15/8	16/11
Antibody titer	log titer	1.45 ± 0.27	1.45 ± 0.28	1.46 ± 0.24	1.45 ± 0.21	1.41 ± 0.07
	GMT (95% CI)	28 (25, 32)	28 (25, 33)	29 (25. 32)	28 (25, 31)	26 (25, 27)
	Number of subjects with titer < 50.3	80	80	77	77	84
Neutralizing activity	log titer	0.72 ± 0.15	0.76 ± 0.23	0.73 ± 0.14	0.73 ± 0.19	0.73 ± 0.13
	GMT (95% CI)	5.3 (4.9, 5.7)	5.8 (5.1, 6.5)	5.3 (4.9, 5.7)	5.4 (4.9, 5.9)	5.3 (5.0, 5.7)

*n*, number of subjects, CI, confidence interval.

**Table 2 vaccines-11-01535-t002:** Treatment-Emergent Adverse Events reported in more than 5% of subjects—Vaccination and Observation Periods (Safety Analysis Set).

	AG0302-COVID19 Intramuscular							
	A (2 mg, Three Times at 2-Week Intervals) N = 85		B (4 mg, Twice at 4-Week Intervals) N = 84		C (8 mg, Twice at 4-Week Intervals) N = 83		Intramuscular N = 252	
	*n* (%)	95% CI	*n* (%)	95% CI	*n* (%)	95% CI	*n* (%)	95% CI
Treatment-emergent adverse events	69 (81.2)	[71.2, 88.8]	74 (88.1)	[79.2, 94.1]	73 (88.0)	[79.0, 94.1]	216 (85.7)	[80.8, 89.8]
Preferred Term								
Diarrhea	5 (5.9)	[1.9, 13.2]	8 (9.5)	[4.2, 17.9]	5 (6.0)	[2.0, 13.5]	18 (7.1)	[4.3, 11.1]
Malaise	20 (23.5)	[15.0, 34.0]	17 (20.2)	[12.3, 30.4]	16 (19.3)	[11.4, 29.4]	53 (21.0)	[16.2, 26.6]
Vaccination site induration	3 (3.5)	[0.7, 10.0]	14 (16.7)	[9.4, 26.4]	12 (14.5)	[7.7, 23.9]	29 (11.5)	[7.8, 16.1]
Vaccination site pain	62 (72.9)	[62.2, 82.0]	71 (84.5)	[75.0, 91.5]	64 (77.1)	[66.6, 85.6]	197 (78.2)	[72.6, 83.1]
Headache	18 (21.2)	[13.1, 31.4]	19 (22.6)	[14.2, 33.0]	15 (18.1)	[10.5, 28.0]	52 (20.6)	[15.8, 26.2]
	**AG0302-COVID19 Intradermal**							
	**D** **(1 mg, Three Times at 2-Week Intervals)** **N** **= 84**			**E** **(1 mg, Twice at 4-Week Intervals)** **N = 86**			**Intradermal** **N = 170**	
	***n*(%)**		**95% CI**	***n*(%)**		**95% CI**	***n*(%)**	**95% CI**
Treatment-emergent adverse events	71 (84.5)		[75.0, 91.5]	73 (84.9)		[75.5, 91.7]	144 (84.7)	[78.4, 89.8]
Preferred Term								
Diarrhea	11 (13.1)		[6.7, 22.2]	20 (23.3)		[14.8, 33.6]	31 (18.2)	[12.7, 24.9]
Malaise	26 (31.0)		[21.3, 42.0]	26 (30.2)		[20.8, 41.1]	52 (30.6)	[23.8, 38.1]
Vaccination site erythema	32 (38.1)		[27.7, 49.3]	23 (26.7)		[17.8, 37.4]	55 (32.4)	[25.4, 39.9]
Vaccination site induration	24 (28.6)		[19.2, 39.5]	30 (34.9)		[24.9, 45.9]	54 (31.8)	[24.8, 39.3]
Vaccination site pain	34 (40.5)		[29.9, 51.7]	35 (40.7)		[30.2, 51.8]	69 (40.6)	[33.1, 48.4]
Vaccination site pruritus	13 (15.5)		[8.5, 25.0]	22 (25.6)		[16.8, 36.1]	35 (20.6)	[14.8, 27.5]
Vaccination site swelling	19 (22.6)		[14.2, 33.0]	16 (18.6)		[11.0, 28.4]	35 (20.6)	[14.8, 27.5]
Headache	21 (25.0)		[16.2, 35.6]	24 (27.9)		[18.8, 38.6]	45 (26.5)	[20.0, 33.8]

## Data Availability

Data will be made available upon reasonable request.

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
