# Peer review of "Long Term Follow-Up Study of a Randomized, Open-Label, Uncontrolled, Phase I/II Study to Assess the Safety and Immunogenicity of Intramuscular and Intradermal Doses of COVID-19 DNA Vaccine (AG0302-COVID19)"

_vaccines, 2023, doi:10.3390/vaccines11101535_

Round 1
Reviewer 1 Report
This is a well designed and executed trial that deserve publicastion. The study is very informative about the potentialities and limitations of DNA vaccination in the field of CoV prevention and informs about the current status of the DNA vaccine technology, specifically the limitation in the induction of Ab responses. The readers get the message from the authors: the technology under study needs further improvement before starting phase III clinical development in future.
One recommendation to authors is to insert a paragraph discussing the AE induced with the DNA vaccine comparing the obtained results with the AE induced by the other vaccine technologies developed against Covid19.
Another recommendation to authors is to modify the conclusions in order to adjust to the declared objectives of assessing immunogenicity, comparing routes etc. At present, the conclusion are mainly comments after a very short and definitive sentence that prevent the focus of the section. In this sense the text should be modified focusing on the objective of the study.
Tables, in page 5 and subsequently, should be organized before final edition in order to avoid the unnecessary confusion due to the selection of the text justification, maybe it would be better to arrange left all the text.
As DNA vaccine technology is suitable for T cell immunity induction, I would recommend that future studies may explore the progression to severe disease in vaccinated vs non vaccinated concommitant group of patients considering hospitalization, progression to severe disease and deaths. If there is information about this from the present vaccinated groups, it would be valuable to include one or two sentences in the Discussion section.
The value of this type of product (DNA Vaccines) may reside in their capacity to induce T cell based immunity, and not preventing infection but preventing disease progression -more than simply focusing on the Ab titers. In the future we may be confronting highly transmissible and more pathogenic variants. The development of a technologies capable of inducing T cell immune responses maz be a suitable way of preventing the progression to severe disease. This type of product may be needed in future considering the high transmissibility of each new variant and the timing to introduce variant-updated vaccines.
In this sense, my recommendation to the journal is to publish the present work after the recommended revision.
Author Response
- One recommendation to authors is to insert a paragraph discussing the AE induced with the DNA vaccine comparing the obtained results with the AE induced by the other vaccine technologies developed against Covid19.
We added the paragraph regarding AEs as follows:
Line 310-314
As for safety profile, both local and systemic adverse events observed in this study were reported with other different types of vaccines, and no adverse events that were considered unique to this vaccine were reported. Although this is a rough comparison, it seemed that the incidence and severity of systemic adverse events such as fatigue, headache, fever, and chills with this DNA vaccine was lower than that of mRNA and vector-type vaccines [3,4,6], and does not much differ from that of the inactivated vaccine [5].
- Another recommendation to authors is to modify the conclusions in order to adjust to the declared objectives of assessing immunogenicity, comparing routes etc. At present, the conclusion are mainly comments after a very short and definitive sentence that prevent the focus of the section. In this sense the text should be modified focusing on the objective of the study.
We modified the conclusion to adjust the objectives as follows:
Line 337-339
Both Intramuscular and intradermal injections of AG0302-COVID19 induced T cell immunity, but AG0302-COVID19 could not elicit sufficient humoral immunity for transition to Phase 3 in this study. There found no particular concerns about side effects.
- Tables, in page 5 and subsequently, should be organized before final edition in order to avoid the unnecessary confusion due to the selection of the text justification, maybe it would be better to arrange left all the text.
We newly prepared two Box-Whisker plot figures showing time-course of antibody titer and neutralizing activity in Figure3 and 4.
- As DNA vaccine technology is suitable for T cell immunity induction, I would recommend that future studies may explore the progression to severe disease in vaccinated vs non vaccinated concommitant group of patients considering hospitalization, progression to severe disease and deaths. If there is information about this from the present vaccinated groups, it would be valuable to include one or two sentences in the Discussion section.
Thank you for giving us very important suggestions. We added the comment regarding the possibility that DNA vaccine may become useful for preventing the progression to severe disease by referencing the representative paper, and added following senses in Discussion.
Line 292-297
Bange et. al reported that COVID-19 patients with hematologic cancer who has COVID-19-specific INF-gamma response showed improved mortality even in limited humoral immunity condition, and they pointed out the importance of CD8 T cell response to vaccination. In future, we may be confronting highly transmissible and more pathogenic variants, therefore, the development of a technologies capable of inducing T cell immune responses based on DNA vaccine may be a suitable way of preventing the progression to severe disease and death.
Reviewer 2 Report
This manuscript delineates follow-up a bit of long-term in phase I/II. The outcomes were not satisfied and could not transferred to phase II, but publishing as a piece or vaccine development would be great. This result may be useful in the future for further vaccine development.
The outcomes in this manuscript were fine and sound. I recommend revising the presentation to make it more attractive to read.
Major concerns.
1. The IRB approved this study, but no IRB approval number(s) are shown in this manuscript.
Suggests adding the IRB approval number(s) to the manuscript.
2. Suggest creating a table delineating the immune response in each timepoint (mimicking Figure 1 but showing all types of antibodies and IFN that you used) to make your data easy to compare. Moreover, suggest creating a graph like Figure 1 with the data set of antibodies.
Minor concerns.
1. Line 44 suggests adding reference, https://www.who.int/news/item/30-01-2020-statement-on-the-second-meeting-of-the-international-health-regulations-(2005)-emergency-committee-regarding-the-outbreak-of-novel-coronavirus-(2019-ncov)
2. Suggest creating a CONSORT scheme for participant enrolment until the endpoint to make it easy to read and follow.
Comments.
1. Line 92 "≥37.5 degrees Celsius ". Suggest using "≥37.5 °C" to shorten this term.
2. Suggest using a subscript form of 50 in the term "ID50" throughout the manuscript.
Typos.
1. Line 351 suggests using "Boehringer Ingelheim" instead of "Boeringher Ingelheim ".
2. Suggest using "COVID-19" instead of "Covid-19"; this term is an abbreviation, and all letters must be in capital letters
Suggest revise typos or some minor error in this manuscript.
Author Response
- The IRB approved this study, but no IRB approval number(s) are shown in this manuscript.
Suggests adding the IRB approval number(s) to the manuscript.
We added the IRB approval number and approval date as follows:
Line 354-360
This study has been approved by the institutional review boards of all study sites: Medical Corporation Heishinkai OPHAC Hospital (Osaka, Approved No. and Date: No. 1115PB on June 18, 2021), Medical Corporation Heishinkai OCROM Clinic (Osaka, Approved No. and Date: No. 1115PB OCROM Clinic on June 18, 2021), Medical Corporation Heishinkai ToCROM Clinic (Tokyo, Approved No. and Date: No. 1115PB ToCROM clinic on June 18, 2021), IUHW Narita Hospital (Chiba, Approved No. and Date: No. FN-1-2104-082 on June 24, 2021), Medical Corporation Shinanokai Shinanozaka Clinic (Tokyo, Approved No. 741P I II on June 23, 2021), Sekino Clinical Pharmacology Clinic (Tokyo, Approved No. and Date: No. 18102082 on July 8, 2021).
- Suggest creating a table delineating the immune response in each timepoint (mimicking Figure 1 but showing all types of antibodies and IFN that you used) to make your data easy to compare. Moreover, suggest creating a graph like Figure 1 with the data set of antibodies.
We added the 2 figures regarding the immune response in each timepoint at line 229-231, and 248-250.
We newly prepared two Box-Whisker plot figures showing time-course of antibody titer and neutralizing.
Minor concerns.
- Line 44 suggests adding reference, https://www.who.int/news/item/30-01-2020-statement-on-the-second-meeting-of-the-international-health-regulations-(2005)-emergency-committee-regarding-the-outbreak-of-novel-coronavirus-(2019-ncov)
We inserted this reference at Line 44 as Reference No.2. The following reference number were changed.
- Suggest creating a CONSORT scheme for participant enrolment until the endpoint to make it easy to read and follow.
We newly created Figure 1 as a CONSORT flow diagram at Line 178-180.
Comments.
- Line 92 "≥37.5 degrees Celsius ". Suggest using "≥37.5 °C" to shorten this term.
We corrected as suggestion in Line 84 and 85.
- Suggest using a subscript form of 50 in the term "ID50" throughout the manuscript.
We corrected as suggestion in Line 115, 119, 120, 236, 238, 241, and 242.
Typos.
- Line 351 suggests using "Boehringer Ingelheim" instead of "Boeringher Ingelheim ".
We corrected as suggestion in Line 373 and 377.
- Suggest using "COVID-19" instead of "Covid-19"; this term is an abbreviation, and all letters must be in capital letters
We corrected as suggestion in Line 156.
Reviewer 3 Report
The authors have proposed a region specific need for the development of DNA based vaccines of, which there is already evidence that a DNA based vaccine is in the market for use. The study does provide a basis for what does not work in the DNA vaccines space.
Minor comments
1. The authors should discuss more on why the route or the method of delivery or dose didn't have any differential effect on the antibody titre post vaccination
Author Response
- The authors should discuss more on why the route or the method of delivery or dose didn't have any differential effect on the antibody titre post vaccination.
We inserted the discussion regarding why the route or the method of delivery or dose didn't have any differential effect on the antibody titer post vaccination in Line 304-309.
Dose independency in production of IFN-γ- cells observed in this study might be partly affected by this inefficient penetration into cell nuclei. In addition, the route or the method of delivery didn't have any differential effect on the antibody titers post vaccination between IM and ID vaccination, suggesting no apparent difference in the sequential antigen synthesis process (i.e., transfection of plasmid DNA into antigen presenting cells, endogenous genes expression, antigen presentation via major histocompatibility in dendric cells, and T-cell responses).